# Fully Automatic Segmentation of Sphenoid Sinus in CT Images with 3D Convolutional Neural Networks

**Kamal Souadih**
Medical Computing Laboratory (LIMED),
University of Abderrahmane Mira,
06000, Bejaia, Algeria
kamal.souadih@univ-bejaia.dz

**Ahror Belaid**
Medical Computing Laboratory (LIMED),
University of Abderrahmane Mira,
06000, Bejaia, Algeria
ahror.belaid@univ-bejaia.dz

**Douraied Ben Salem**
INSERM UMR 1101
Laboratory of Medical Information Processing (LaTIM),
5 avenue Foch, 29200 Brest, France,
Neuroradiology and Forensic Imaging Department,
CHRU Brest, La Cavale Blanche Hospital. Boulevard Tanguy Prigent, 29609 Brest, France,
douraied.bensalem@chu.brest.fr

## Abstract

Today, Deep learning algorithms have quickly become essential in the field of medical image analysis. Compared to the traditional methods, these Deep learning techniques are more efficient in extracting compact information leading towards significant improvement performance of medical image analysis system. We present in this paper a new technique for sphenoid sinus automatic segmentation using a 3D Convolutional Neural Networks (CNN). Due to the scarcity of medical data, we chose to used a 3D CNN model learned on a small training set. Mathematical morphology operations are then used to automatically detect and segment the region of interest. Our proposed method is tested and compared with a semi-automatic method and manual delineations made by a specialist. The preliminary results from the Computed Tomography (CT) volumes seem to be very promising.

## 1 Introduction

The sinuses anatomy in general is very complex and variable [1]. Sphenoid sinus is too, a very variable cavity, an important landmark in surgery and at the same time it is hard to isolate [2-3-4]. Fig. 1 shows a diagrammatic representation of the paranasal sinuses location. Another difficulty is that the sinuses can also be divided into many nooks, which communicate with each other through an incomplete bone wall [5], which further complicates their localization, see for e.g. [5]. The complications while operating on sphenoid sinus are easily avoided if we know its anatomical features [6].

As it has been established, the sphenoid sinus is the most inaccessible part of the face, being inside the sphenoid bone and involving a number of different structures. Its deep anatomical location makes it difficult to approach. This deep location can be beneficial in the case of forensic identification. Unlike other sinuses, the sphenoid sinus is well protected from traumatic degradation resulting from external causes.

sphenoid sinuses can be classified according to their positions in the sella turcica into four types [7]:

- Conchal: complete missing or minimal sphenoid sinus;

1st Conference on Medical Imaging with Deep Learning (MIDL 2018), Amsterdam, The Netherlands.

- Pre-sellar: the posterior wall of sphenoid sinus is in front of the anterior wall of the sella turcica
- Sellar: the posterior wall of the sphenoid sinus is between the anterior and posterior walls of sella turcica
- Post-sellar: the posterior wall of sphenoid sinus is behind the posterior wall of the sella turcica

These types of sphenoid sinuses and their basic dimensions (height, width and depth) can generally help predict the risk of accidental injury, but also useful for individual identification as can be seen in [8].

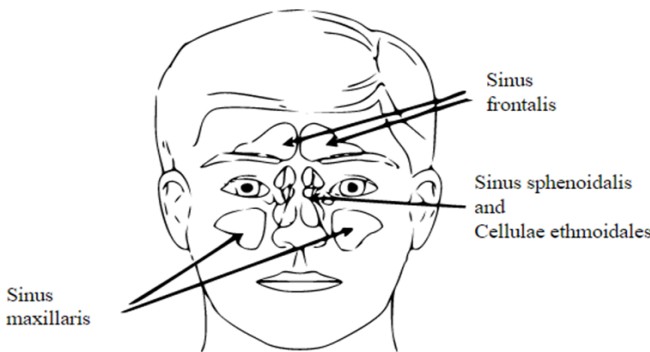

Figure 1: Diagrammatic representation of paranasal sinuses.

The computed tomography (CT) is an excellent imaging method used for the assessment of sinuses anatomy as it allows precise evaluation craniofacial of bones and the extent of there pneumatization [9-4]. By using segmentation of three dimensional (3D) CT-images of sphenoid sinus we could make useful measurements of its volume anatomy [10].

A 3D segmentation is a technique that consist labeling each voxel in an image and assigning it in group of voxels that define an anatomical structure. This technique has a wide variety of applications in medical research and Computer-Aided Diagnosis.It is a very useful method, it allows to extract and recognize organs like: the heart, the brain, the spine, the blood vessels,etc.It is used too to improve visualization of medical images and allow quantitative measurements of organs structures on the image. Segmentation is also important in building anatomical atlases, researching shapes of anatomical structures and tracking their changes over time [11].

The artificial intelligence techniques represented by machine learning are increasingly used in medical image analysis and segmentation. In the recent years, the appearance of deep learning techniques has contributed significantly improving for medical image analysis, based on convolutional neural networks (CNN) that give the ability to automatically learn significant patterns and extract real structures from images [3-12].

One of the main reasons for the success of the CNN model was that it possible to directly use the pre-trained model to do various other tasks which it was not originally intended for. It became remarkably easy to download a learned model, and then tweak it slightly to suit the application at hand [13]. To the best of our knowledge, there is no an automatic segmentation approach dedicated to sphenoid sinuses. This is probably due to the complex anatomy and high anatomical variability. Another particular challenge we had to overcome includes the opening of sphenoid sinus ostium, making the wall delimitation very difficult. In this paper, we made a sphenoid sinus automatic segmentation tool that uses conventional CT-images based on 3D CNN. The proposed method is efficient, robust, and is able to obtain good results with small training dataset.

## 2 Method

Our automatic sphenoid sinus segmentation method consists of three main steps, where the result of step is the input of another one. The first step is a preprocessing step; we create and transform

automatically the images volume given from a PACS to an image of the region of interest. Than we perform a segmentation with 3D deep CNN [14], that we adapted and parameterized to produce highly accurate sinus segmentation. Finally a postprocessing based on mathematical morphology operations is carry out sinus measurement and refine segmentation (Figure.1). This splitting in stages allowed us to improve and simplify the use of CNN at the CPU level. In the following we describe the method stage:

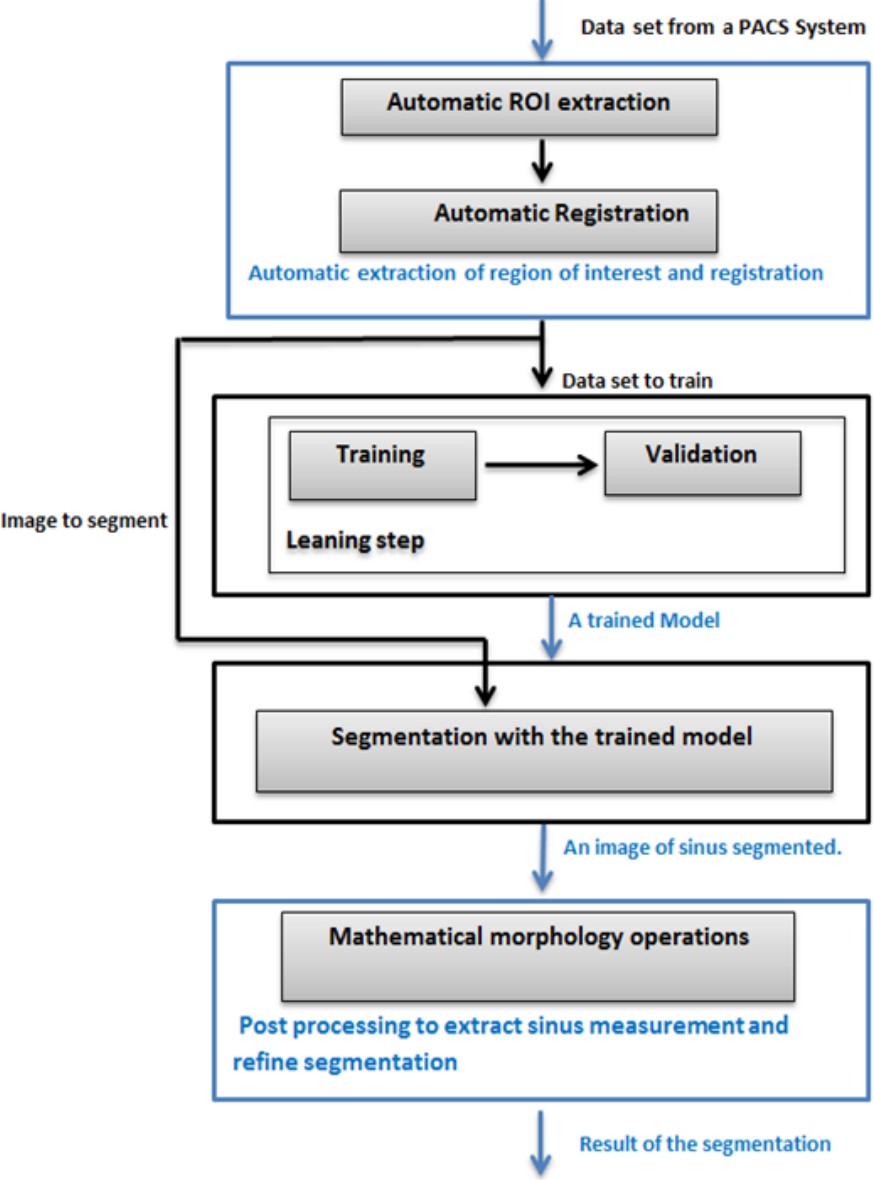

Figure 2: Flow chart of the sphenoid sinus segmentation scheme.

## 2.1 The automatic ROI extraction for the CT-image:

The preprocessing step uses some interesting techniques with slight transformations that are adapted to improve the effectiveness of the specific type of segmentation method used in the next step. These transformations are made so that common parameters can be used for all images of all intensity ranges. In other words, we aim to operate only on a reduced 3D region, a region of interest centered on the sinus at issue and not on the whole image. This region of interest must be the same in terms

of dimensions for all images in data set of training or test. To achieve this, we first selected a target image with a well-oriented head and a clear sinus.

We manually traced a large rectangle, enough to contain the sinus whatever its shape,size does not exceed 200 x 200 x 200 pixels. This rectangle will also serve as a reference bounding-box. Then, all other database images are registered onto this target image with its bounding-box. As the images are coming from different persons, we choose to use a rigid registration, allowing a correction of the different positions and orientations arising from the clinical exam. Since the natural size of the skulls is different from one person to another, we have avoided using affine registration[15], which risks distorting the estimation volume that will be used later as parameter for identification. Thereby, we were able to build a new database consisting only of regions of interest, with the same size as the reference box.

## 2.2 Sinus Segmentation with deep 3D CNNs

In this step we employ DeepMedic [16] realized as an open source software [17], it is an architecture with adjustable number of deep layers, double-pathway and 3D Convolutional Neural Network, developed for the segmentation of brain lesions [14]. This system segments MRI 3D images corresponding to a multi-modal 3D patch at multiple scales. For our study we used the lightweight version CPU-based of this software to drive our sinus automatic segmentation model; in our case we use one modality and a CT images format. This CPU model gives a satisfactory solution to our problem.

The robustness of this CNN was tested when less training data were available or fewer filters were used, this architecture was further benchmarked on the BRATS 2016 Challenge, where it achieved very good performance despite the simplicity of the pipeline [17]. It was demonstrated that it is possible to train this 3D CNN on a small dataset of 28 cases. This network was given a good result on the task of segmenting ischemic stroke lesions, accomplishing a mean Dice of 64% (66% after post processing) on the ISLES 2015 training dataset, ranking among the top entries [14]. This architecture[16]based on :

- Two parallel convolutional pathways that process the input at multiple scales to achieve a large receptive field for the final classification while keeping the computational cost low.
- A small convolutional kernels. That gives efficiency to building deeper CNNs without severely increasing the number of trainable parameters and Inspired by VGG (Very deep convolutional networks)[18].Building high performing and efficient 3D CNNs thanks to the much smaller computation required for the convolution with small $3^3 kernels$.
- Full convolutional fashion on image segments in both training and testing stage.

In what follows we will present the main algorithms that make up this architecture, In [14] the creators and authors of this architecture presented a very clear and detailed of DeepMedic architecture with its theoretical background, here we just giving a summary of each step, which make up this software:
1- Each layer $l \in [1, L]$ consists of $C_l \ feature \ maps \ (FM)$ also referred to as $Channels$
2- Every $FM$ represents a group of neurons that detect a particular pattern (a feature, in the channels of the previous layer).
3- A $Pattern$ is defined by kernel weights associated with the $FM$
4- If the neurons of the $m_{th} \ FM$ in the $l_{th}$ layer are arranged in a 3D grid, their activations constitute the image defined bye:

$$y_l^m = f \left( \sum_{n=1}^{c_{l-1}} k_l^{m,n} \right) * y_{l-1}^n + b_l^m.$$

- $y_l^m$ is the result of convolving each of the previous layer channels with a 3-dimensional.
- $k_l^{m,n}$ Is a $kernel$ , adding a learned $bias \ b_l^m$ applying a non-linearity $f$
- The image $y_0^n$ is the input to the first layer, correspond to the channels of the original input image.

5- Each kernel is a matrix of learned hidden weights $W_l^{m,n}$
6- Each $class$ of segments has a $C_l$ number of .

7- The activations of $C_l$ are fed into a position-wise $softmax$ function that produces the predicted posterior

$$p_c = \exp\left(y_L^c\left(X\right)\right) / \sum_{c=1}^{C_L} \mathbf{exp}\left(y_L^c\right).$$

- $y_L^c$ is the activation of the FM at position $l \in N^3$

8- The size of the neighbourhood of voxels $\varphi_l$ in the input that influence the activation of a neuron is a receptive field, increases at each subsequent layer and is given by the 3-dimensional vector:

$$\varphi_l^{\{x,y,z\}} = \varphi_{l-1}^{\{x,y,z\}} + \left(k_l^{\{x,y,z\}} - 1\right)\tau_l^{\{x,y,z\}}. \qquad (1)$$

Where

- $k_l, \tau_l \in N^3$ are vectors expressing the size of the kernels and stride of the receptive field at layer $l$
- $\tau_l$ is given by the product of the strides of kernels in layers preceding, in this system the $\tau_l = (1,1,1)$
- $\varphi_{CNN} = \varphi_L$ : This is called theCNN's receptive field; the receptive field of a neuron in the classification layer corresponds to the image patch that influences the prediction for its central voxel.

9- The dimensions of the FMs in Layer $l$ is given by:

$$\delta_l^{\{x,y,z\}} = \left\lceil \frac{\delta_l^{\{x,y,z\}} - \varphi_l^{\{x,y,z\}}}{\tau_l^{\{x,y,z\}}} + 1 \right\rceil. \qquad (2)$$

10- If an input of size $\delta_{in}$ is provided, $\delta_{in} = \varphi_{CNN}$ is a size of input patch in the common patch-wise. The FMs of this classification layer have $1^3$.

11- CNNs are trained patch-by-patch and random patches of size $\varphi_{CNN}$ are extracted from the training images.

12- To maximize the log likelihood of the data or, equally, minimize the Cross Entropy via the cost function is used:

$$J\left(\theta; I^i; C^i\right) = -1/B \sum_{i=1}^{B} \log(P(Y = c^i | I^i, \theta)) = -1/B \sum_{i=1}^{B} \log p_{C^i}. \qquad (3)$$

- $B$ is the size of batch, which is then processed by the network for one training iteration of Stochastic Gradient Descent (SGD).
- The pair $(I^i, C^i), \forall i \in [1, B]$ is the $i_{th}$ patch in the batch and the true label of its central voxel.
- The scalar $p_{C^i}$ is the predicted posterior for Class $C^i$
- Regularization terms were omitted for simplicity. Multiple $C^i$ Sequential optimization steps over different batches gradually lead to convergence.

13- The classification layer is the activation of the last layer of CNN.

Memory requirements and computing time increase with batch size, which is the limitation of 3D CNNs, DeepMedic uses a strategy that exploits the dense inference technique on image segments. Following from Eq.(2), if an image segment of size greater than $\varphi_{CNN}$ is given as input to the network, the output is a posterior probability for multiple voxels $V = \prod_{i=\{x,y,z\}} \delta_l^i$. If the training batches are formed of $B$ segments extracted from the training images, the cost function Eq.(3), in the case of dense-training[14] becomes:

$$J_D\ (\theta; I_s; C_s) = -\frac{1}{B*V}\ \sum_{s=1}^{B}\sum_{v=1}^{V} p_{c_s^v}\ (x^v)\,. \qquad (4)$$

Where $I_s$ and $C_s$ are the $s_{-th}$ segment of the batch and the true labels of its $v_{-th}$ voxel. $x^v$ the corresponding position in the classification FMs and $p_{c_s^v}$ the output of the softmax function. The effective lot size is increased by a factor V without corresponding increase in calculation and memory requirements DeepMedic architecture is also a deep architecture based on small $3^3$ kernels that are faster to convolve with and contain less weights[14].

We have adapted the 3D CNN for five layers, with a receptive field of size $17^3$ and one modality. The classification layer (the last layer) is implemented like a convolutional with $1^3$ kernels, which enables efficient dense inference. When the network segments an input it predicts multiple voxels simultaneously, one for each shift of its receptive field over the input (see Figure 4). The training time required for convergence of the final system is roughly 20 minutes using a CPU Intel I5-7300 with 2x2.5 GHz. Segmentation of a 3D scan of a sphenoid sinus requires 1 minute.

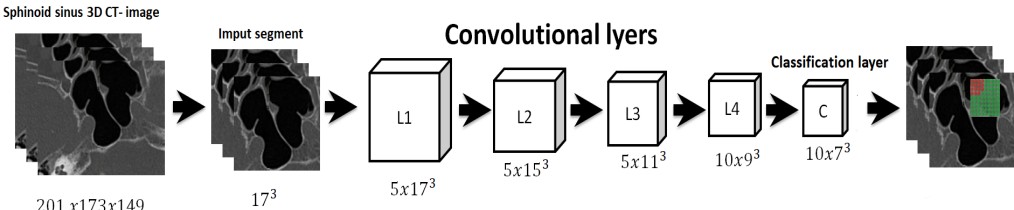

Figure 3: Architecture of the deepMedic for automatic sphenoid sinus segmentation.

## 2.3   Post processing

The segmentation result obtained by the 3D CNN of the precedente step method does not make it possible to distinguish between the sphenoid sinus from the other sinuses. The nasal cavities as well as the paranasal sinuses have almost the same gray level intensity. To differentiate the sinuses, we have used a prior knowledge about the positioning of these sinuses. Indeed, the sphenoid sinus is the deepest cavity starting from the front face, and therefore it is the first cavity encountered from the back of the skull at the median. Thus, using the operations of mathematical morphology we have been able to locate the sphenoid sinus. We have first applied an erosion operation to the segmented image which allows removing the residues, but especially the potential connections between the sphenoid sinus and other cavities. More precisely, erosion operation allows to remove the ostium and to well separate the two hemisinus of the sphenoid sinus.

Once the sphenoid sinus cleared, we have subsequently calculated the centres of gravity of all the regions on the image. After sorting the centers coordinates along the coronal axis, the deepest centre corresponds, of course, to the region of the sphenoid sinus, or more precisely corresponds to the deepest hemisphere. When the hemisphere is segmented from the rest of the cavities, a dilation operation (with the same parameters as the previews erosion) is applied to recover some details of the shape lost during erosion operation. As can be seen, the detection of the two hemispheres of the sinus is sequential. Indeed, after removing the first detected hemisphere, the same process is launched on the initially segmented image.

## 3   Result

### 3.1   Dataset

Our dataset has 24 Head CT images,which were performed on a helical, multi-detector CT scanner.Some data exclusion criteria have been set. All CT exam with head fractures, tumors, or any pathological process involving the sphenoid bone and the surrounding structures, but also with sinuses mucosa thickening, or any abnormality of the sinuses contents, were not included in the study. After the preprocessing we have obtained 3D CT- images less than 200 x 200 x 200. We have used 15

Table 1: Elements of our dataset

| Data set | Number of images |
| --- | --- |
| Total CT exam considered | 24 |
| Total CT exam on train step in the 3D CNN Algorithm | 5 |
| Total CT exam on validation step in the 3D CNN Algorithm | 10 |
| Total images for test step (automatic segmented) | 9 |
| Total CT exam manually segmented by an expert assistance for train step | 5 |
| Total CT exam manually segmented by an expert assistance for validation step | 10 |

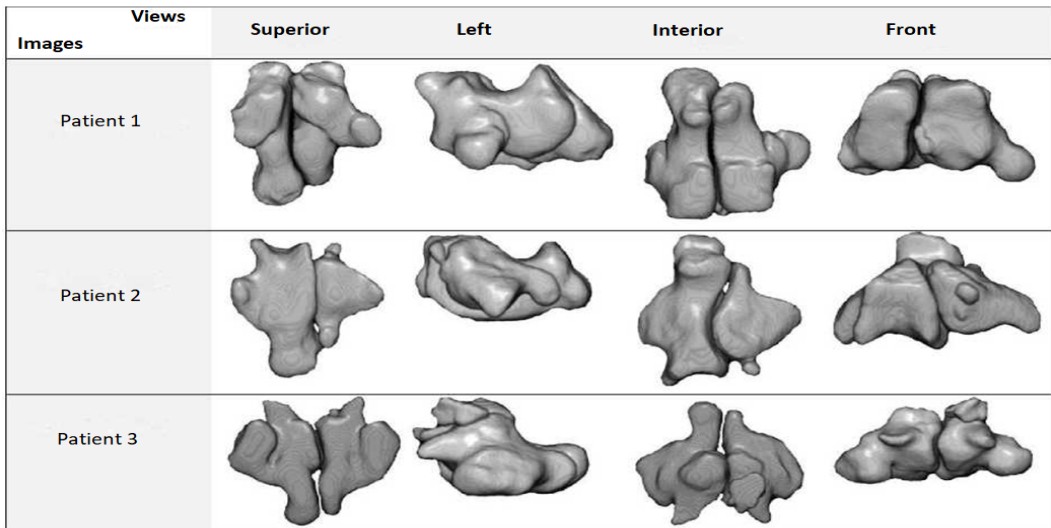

Figure 4: Segmentation examples for 3 CT-images, shows a superior, left, interior and front views.

images for training step (training and validation) in the 3D CNN algorithm and 9 images to test. The training dataset need a manual segmentation of spheroid sinus for each image, so we did this manual segmentation assisted by a radiologist, a description of the dataset images used in the 3D CNN Algorithm is illustrated and summarized in Table 1.

### 3.2 Results

An example of 3 segmentations is reported in Figure 4. It shows the result of the segmentation and the extracted a sphenoid sinus as explained in the previous sections. The segmentation is performed using the 3D CNN and affine with the morphological operations.

### 3.3 Validation

To evaluate the accuracy and robustness of the proposed automated approach, the results from the same 9 sphenoid sinus automatically segmented with our tool were compared with a semi-automatic Clustering method segmentation of ITK-SNAP Software with a manual segmentation that was performed with an experienced radiologist using a standard procedure. Each image was segmented by carefully tracing the outlines of the sphenoid sinus while following the inner bone surface, proceeding in an axial direction. An example of the Spheroid sinus manual segmentation process of one slice is shown in Figure 5.

The Dice Similarity Coefficient (DSC), Hausdorff distance (HD) and Mean Absolute Distance (MAD), were used for evaluating the proposed method. The dice Coefficient (DSC), one of the most common methods for evaluating segmentation results, indicates a level of similarity between the reference (manual segmentation) and segmented result (automatic segmentation), the formulation of DSC is given by:

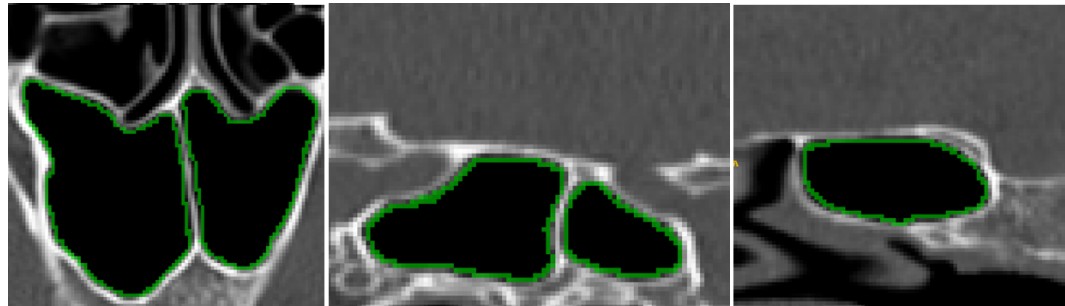

Figure 5: Example of the process of manual segmentation on one slice. From left to right: an axial, sagittal and coronal view.

$$DSC = \frac{2N(S_1 \cap S_2)}{N(S_1) + N(S_2)}. \qquad (5)$$

Where $S_1$ and $S_2$ represent the obtained segmentation and the ground truth respectively (manual segmentation), and $N(.)$ defines the number of pixels.

$$DCS \in [0, 1]$$

the closer the DCS value to 1, the better the segmentation is. The Hausdorff distance is metric represents the spatial distance between two point sets, i.e., is the maximum distance between two point sets $C1$ and $C2$, from each point $a \in C1$ to point $b \in C2$ and vice versa. HD is defined as follows:

$$HD(C_1, C_2) = \max(h(C_1, C_2), h(C_2, C_2)). \qquad (6)$$

The Mean Absolute Distance (MAD) metric . Is given as follows:

$$MAD(C_1, C_2) = \frac{1}{2}\left[\frac{1}{n}\sum_{i=1}^{n} d(a_i, c_2) + \frac{1}{m}\sum_{j=1}^{m} d(bj, c_1)\right].$$

Where the distance between the point ai and the closet point $b_j$ is given by :

$$d(a_i, c_2) = min\|b_j - a_i\|.$$

Where $b_j \in C_2$.

The three metrics: DSC, HD and MAD were measured for all segmentations; Tables 2 and 3 illustrate the associated results and a comparison between our automatic segmentation and semi-automatic clustering of ITK-SNAP for the nine CT- images respectively with a manual segmentation. Related mean, median and standard deviation are shown in the same tables.

Table 2: Comparison results with manual delineations. Are shown, DSC, Hausdorff (HD) and MAD measures for proposed approach and semi-automatic method using ITK-SNAP.

| Measures | DSC (%) | | | HD (mm) | | | MAD (mm) | | |
|---|---|---|---|---|---|---|---|---|---|
| Index | Mean | Median | SD | Mean | Median | SD | Mean | Median | SD |
| Our tool | 95.81 | 96.16 | 1.48 | 9.87 | 8.39 | 5.31 | 3.24 | 2.22 | 2.20 |
| ITK-SNAP | 96.01 | 95.94 | 0.54 | 9.23 | 8.19 | 5.30 | 3.16 | 2.10 | 2.20 |

## 4 Discussion and conclusion

To our knowledge, only manual or semi-automatic methods have been applied for sphenoid sinus segmentation. These techniques present inter- and intra-observer variability and are both time-consuming. In the present work, we have developed a fully automated method for sinus sphenoid segmentation

Table 3: Detailed results of comparison between the proposed automatic and semi-automatic (ITK-SNAP) segmentation for 9 volumes, using respectively DSC, HD and MAD distances.

| CT Volumes | 1 | 2 | 3 | 4 | 5 | 6 | 7 | 8 | 9 |
|---|---|---|---|---|---|---|---|---|---|
| Our results | 96.52 | 96.10 | 95.59 | 92.10 | 95.71 | 96.84 | 95.91 | 95.48 | 96.87 |
| ITK-SNAP results | 95.78 | 96.10 | 95.74 | 96.74 | 95.21 | 97.15 | 96.16 | 95.77 | 96.33 |
| Our results | 4.09 | 7.02 | 6.62 | 2.71 | 0.96 | 2.08 | 1.58 | 1.85 | 2.22 |
| ITK-SNAP results | 4.08 | 6.88 | 6.61 | 2.06 | 0.93 | 2.10 | 1.54 | 2.01 | 2.20 |
| Our results | 10.43 | 15.43 | 19.46 | 13.32 | 3.79 | 8.07 | 4.76 | 5.15 | 8.39 |
| ITK-SNAP results | 43.76 | 15.43 | 19.46 | 6.29 | 3.87 | 8.19 | 4.67 | 5.18 | 8.41 |

based on CT images. The statistical comparisons between our automated tool segmentation and the clustering ITK-SNAP semi-automatic segmentation with manual segmentation methods revealed strong agreement and low dispersion between variables. These promising findings were maintained over the entire range of sphenoid sinus segmentation evaluation, and the mean difference between the automated and manual techniques was approximately 5% for both measurements. These differences are sufficiently small and this gives us a good confidence in this segmentation method. This automated tool has the ability of segmenting a 3D CT-image in approximatively under then 1 minute. Furthermore, this tool does not require complex or expensive equipment although it uses 3D CNN. This method may be applied using conventional computers, thus allowing better implementation in clinical practice.

The present study has some limitations. Our methodology was analyzed using only one protocol with a slice thickness of 0.5 mm. Prionas et al.[19] reported a greater error of volume quantification for thicker slices. Further studies are needed to evaluate volume in patient groups with different ages, genders, and ethnicities. Nevertheless, the automated tool may be adapted to quantify volume in other paranasal sinuses.

In conclusion, the present study found a good correlation between the manual and automated sphenoidal sinus volume estimation techniques. Our automated measurements of sphenoidal sinus volume based on CT exams were reliable, robust, and accurate compared with the manual method. Our findings suggest that this automated tool may be applied in clinical practice.It does not require substantial user expertise, and it is reproducible and fast.

**Acknowledgments**

The authors would like to thank Rabeh Djabri for English proofreading.

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
