# OpenReview forum: "Fully Automatic Segmentation of Sphenoid Sinus in CT Images with 3D Convolutional Neural Networks"
_MIDL.amsterdam/2018/Conference — Submitted to MIDL 2018_

### Review · AnonReviewer1 · 2018-05-04
**limited contribution, preliminary validation**

**Rating:** 1
**Confidence:** 2

**Review:**

This paper addresses the problem of segmenting sphenoid sinus in CT scans. This is an application paper, where a standard CNN model is applied to perform patch based segmentation, with pre- and post-processing steps. The pre-processing step involves the extraction of a region of interest and a registration step. The post-processing step is composed of mathematical morphology operations.

pros
+ relevant problem
cons
- contribution is limited, validation is preliminary

The main concern based on this paper is related to  contribution and the experimental evaluation.

*Although it might be the case that CNNs have not previously been applied to segment sphenoid sinus, the proposed pipeline has been heavily exploited in the medical imaging literature.
*The paper is longer than what suggested by the conference guidelines. Given that there are a couple of pages detailing how to perform a convolution operation, explaining concepts such as receptive fields and feature maps, it does not seem necessary to go beyond 8 pages.
*Please review all Eq in the paper. The output of a layer seems to sum the kernel values and apply a non-linearity to the sum, prior to the convolution operation. There is no need to have 2 notations for the kernel (k and W).
*The model is trained in a very constrained setting, where data containing fractures and so on has been removed. Why not train the models with all the data you have instead of removing the samples with fractures and so on?
*Table 1: it is not clear whether you actually have expertly labeled segmentation for the test samples.
*Comparison to state-of-the-art literature is very limited, the method is only compared to a semi-automatic toolkit, without highlighting benefits and limitations of each one of those.
*Reported results are preliminary and might constitute a good baseline if properly compared to previous approaches.
*Why not use fully convolutional networks as an alternative segmentation model?
*Why not train the model end to end, including ROI extraction and registration?

**Special Issue:**

No

---

### Review · AnonReviewer2 · 2018-05-10
**Use of fCNN method for a CT segmentation application. No clear contribution or educational value.**

**Rating:** 1
**Confidence:** 3

**Review:**

This paper evaluates the use of an fCNN architecture for sphenoid sinus segmentation in CT. The authors use the DeepMedic framework. However, the multi-scale architecture specific to DeepMedic is not used. After the CNN stage of the method, there is a post-processing step using morphological operations. The method is trained on 15 annotated images and tested on 9 images.  The method is compared against a semi-automatic clustering method in ITK-SNAP. There is no evaluation of different architectural variations of the network or different optimizers. Based on the images shown in the paper (figure 5), I personally believe the segmentation of the sphenoid sinus should be easy for any modern deep learning segmentation method.
I believe that, in the current form, this paper is not interesting for the MIDL community. To improve this paper, I believe the authors should think about what the community can learn from this work. Virtually no one in the community is working on sphenoid sinus segmentation in CT, so the message should be broader than just this application. For example; is there something specifically hard in the data (demonstrate with images) that is similar to other  medical imaging segmentation tasks? And did you come up with a novel solution for that specific problem? Or perhaps, did you find a way to train these kind of networks much faster? Or, can you train them with the same accuracy with much less data? In this way, a paper discussing a very specific application can still be interesting for the community.

**Special Issue:**

No

---

### Review · AnonReviewer3 · 2018-05-10
**The research is not thorough - there are many weaknesses. The paper is not well structured and it is not nicely written. The data is limited.  The problem is interesting.**

**Rating:** 1
**Confidence:** 3

**Review:**

The authors propose 3D convolutional approach for sphenoid sinus segmentation in CT scans. The approach first considers ROI extraction, followed by the 3D Conv algorithm and some preprocessing.
The methodology used is standard and it involves 3D Convolutional networks that have been heavily used for medical image analysis. Moreover, such approaches have been used for segmentation problems. The problem of sphenoid sinus segmentation as a specific one is interesting and using DL for such problem might be novel.
However, the paper has many weaknesses. The paper is badly written. There are many mistakes from language nature from, mistake, unclear sentences, improper capitalization etc. There is also big inconsistency in the equations, especially in the notation. There is also no need to re-write well known convolution operation and how the optimization problem is stated. It also takes more space in comparison to the part that presents the results with the paper contributions.
The method of: ROI filtering, then 3D conv and some preprocessing have been exploited in the literature. It also requires a lot of hand-crafting and uncertainties. Related to the segmentation, there are already end to end deep learning solutions, like V-net or Seg-Net that could have been taken. Ideally, these approaches introduce less bias and they should have been compared with the proposed solution.
The dataset is very small and it is very uncertain whether bold conclusions could be made on such small dataset. Augmentation is not considered, although it is indeed not natural to enlarge the dataset by using standard augmentation procedures like rotations or contrast. More data is needed or smarter augmentation procedures.
The reviewer cannot recommend acceptance. Although the problem is of interest, the research is not properly done, the paper is not nicely written to satisfy the standards of MIDL and the data used is extremely small.

**Special Issue:**

No

---

### Decision · Program_Chairs · 2018-05-15
**Paper21 Acceptance Decision**

Reject